# Prostate-Specific Membrane Antigen (PSMA)-Positive Extracellular Vesicles in Urine—A Potential Liquid Biopsy Strategy for Prostate Cancer Diagnosis?

**DOI:** 10.3390/cancers14122987

**Published:** 2022-06-17

**Authors:** Susann Allelein, Keshia Aerchlimann, Gundula Rösch, Roxana Khajehamiri, Andreas Kölsch, Christian Freese, Dirk Kuhlmeier

**Affiliations:** 1Fraunhofer Institute for Cell Therapy and Immunology (IZI), 04103 Leipzig, Germany; keshia.aerchlimann@izi.fraunhofer.de (K.A.); andreas.koelsch@izi.fraunhofer.de (A.K.); dirk.kuhlmeier@izi.fraunhofer.de (D.K.); 2Fraunhofer Institute for Microengineering and Microsystems (IMM), 55129 Mainz, Germany; gundula.roesch@kgu.de (G.R.); roxana.khajehamiri@outlook.de (R.K.); christian.freese@imm.fraunhofer.de (C.F.)

**Keywords:** extracellular vesicles, prostate-specific membrane antigen, microarray, immunomagnetic isolation, automated, prostate cancer

## Abstract

**Simple Summary:**

Prostate cancer is the second most commonly diagnosed cancer and the fifth leading cause of cancer-related death in men. It is a generally slow-growing cancer that—when detected in its early stages—has high chances of successful treatment. Just like all cells, cancerously degenerated cells release extracellular vesicles (EVs) to communicate with other cells. The aim of our research was to specifically isolate prostate cancer-derived EVs from urine, characterize the EV surface markers of a prostate cancer cohort, and assess the potential value of the prostate-specific membrane antigen (PSMA) as a biomarker for liquid biopsy in early cancer diagnostics. Our findings demonstrate that the automated isolation of EVs allows for an overall improvement of the precision in sample purification in comparison to manual isolation, thus optimizing the further characterization of EV surface markers as well as evaluating their use in clinical application.

**Abstract:**

All cells release extracellular vesicles (EVs) to communicate with adjacent and distant cells. Consequently, circulating EVs are found in all bodily fluids, providing information applicable for liquid biopsy in early cancer diagnosis. Studies observed an overexpression of the membrane-bound prostate-specific membrane antigen (PSMA) on prostate cancer cells. To investigate whether EVs derived from communicating prostate cells allow for reliable conclusions on prostate cancer development, we isolated PSMA-positive, as well as CD9-positive, EVs from cell-free urine with the use of magnetic beads. These populations of EVs were subsequently compared to CD9-positive EVs isolated from female urine in Western blotting, indicating the successful isolation of prostate-derived and ubiquitous EVs, respectively. Furthermore, we developed a device with an adapted protocol that enables an automated immunomagnetic enrichment of EVs of large sample volumes (up to 10 mL), while simultaneously reducing the overall bead loss and hands-on time. With an in-house spotted antibody microarray, we characterized PSMA as well as other EV surface markers of a prostate cohort of 44 urine samples in a more simplified way. In conclusion, the automated and specific enrichment of EVs from urine has a high potential for future diagnostic applications.

## 1. Introduction

Intercellular communication is mediated by extracellular vesicles (EVs), nanosized, membranous particles released by all cells [1]. Based on their biogenesis, EVs are divided into exosomes and microvesicles. They differ in size, morphology, biophysical attributes, and molecular composition, which is most likely to be specific to their origin [2,3]. EVs transfer information via biologically active surface molecules and vesicular cargo (proteins, nucleic acids, lipids) from their origin to adjacent and distant cells. In host defense, EVs release innate immune proteins [4], act as a pro- or anti-inflammatory mediator, and present antigens on their surface or induce an antitumor response [2,5,6]. In cancer, EVs are involved in the establishment and maintenance of the tumor microenvironment via the stimulation of angiogenesis and the reprogramming of metabolic activity [7,8]. Thereby, signaling molecules are transmitted, tumor heterogeneity is extended, and stromal cells are reprogrammed to promote tumor progression [9,10,11]. In contrast, the immune response is downregulated to escape immune surveillance, evade growth suppression, and resist cell death [12,13]. Cancer-derived EVs can circulate to distant cells, directing organotropic behavior in order to initiate a premetastatic niche [14]. They display distinct protein and RNA patterns, e.g., elevated quantities of tumor suppressor miRNAs compared to healthy cell-derived EVs [15]. It has been further shown that cancer patient-derived EVs transfer malignant traits and confer the same phenotype of primary tumors to oncosuppressor-mutated cells [16].

These features, in combination with the fact that EVs are found abundantly in all bodily fluids, qualify EVs as minimally invasive biomarkers for early diagnosis. For the prognosis of the patient, early diagnosis is crucial. In prostate cancer (PCa), the 5-year survival is 98% when detected in an early stage, and it drops down to only 30% when diagnosed at a metastatic stage [17]. However, it remains challenging to evaluate tumor-specific EVs, since only a small fraction derives from tumor cells. As all sorts of cells release EVs, it might be difficult to detect a disease-specific signature when analyzing the entire EV population. Elaborate and time-consuming methods are required to better isolate and analyze tumor-derived EVs from bodily fluids. However, this is in conflict with the clinical requirements that detection methods need to meet in order to find clinical applications. Namely: be as simple as possible, inexpensive, and have a high sample throughput. Prostate cells express the prostate-specific membrane antigen (PSMA) on their surface and an increased expression was detected for PCa. In addition, an adverse relation with the survival of the patients was observed. The higher the PSMA levels, the lower the survival chance of the patient [18].

In this study, PSMA was investigated as a suitable target for urinary EVs (uEVs) derived from communicating prostate(-cancer) cells in order to provide the ability to specifically isolate them from urine samples in a possible clinical setting. This might be used for the effective diagnosis of PCa using, e.g., RNA cargo analysis [19].

## 2. Materials and Methods

### 2.1. Antibodies


cancers-14-02987-t001_Table 1Table 1Antibodies and their clone ID and conjugation used for Western blot (WB), immunomagnetic isolation (IM), or antibody microarray (MA).TargetCloneSupplierApplicationConjugationCD9HI9aBioLegendWB, IM, MAnone, biotin, APCCD9D3H4PCell SignalingTechnologyWBnoneCD63H5C6BioLegendIM, MAnone, biotin, APCCD815A6BioLegendIM, MAnone, biotin, APCPSMALNI-17BioLegendWB, IM, MAnone, biotinPSMAGCP-05Thermo Fisher ScientificMAnonePSMAD4S1FCell Signaling TechnologyWBnoneTSG101EPR7130AbcamWBnoneEpCAMVU-1D9Thermo Fisher ScientificMAnoneROR12A2BioLegendMAnoneCD24M1/69BioLegendMAnoneHer224D2BioLegendMAnoneNCAMMEM-188BioLegendMAnoneNCAM-L15G3eBioscienceMAnoneCA125986808R&D SystemsMAnoneCA72-4CC49OriGeneMAnoneIC moMOPC-21BioLegendWB, MA, IMbiotinanti-mo
DianovaWB, MACy3anti-rb
DianovaWBHRP


### 2.2. Immunomagnetic EV Isolation

Immunomagnetic beads can target disease/tissue-specific EVs. Here, PSMA, as a PCa-specific marker, was approached, and CD9, as a ubiquitous EV surface marker, was used as a positive control next to the isotype control as a negative control. For the manual isolation procedure, Streptavidin MyOne T1 Dynabeads (Thermo Fisher Scientific, Waltham, MA, USA) magnetic beads were washed three times with PBST (PBS (Gibco) with 0.1% (*v*/*v*) Tween 20 (Carl Roth)). Per µL initial bead suspension, 0.25 µg of antibody was added and incubated for 1 h at room temperature under constant agitation. After four PBST washing steps of 200 µL each, the cell-free urine sample with 0.1% (*v*/*v*) Tween 20 was mixed with the antibody-coated beads and incubated for 1 h at RT under rotation at 8 RPM. Tubes were placed into a magnetic rack for 2 min to achieve bead separation from the sample. Unbound material was removed by four repetitive PBST washes. Elution of captured uEVs on the magnetic beads was performed by adding 35 µL of glycine (0.1 M, pH 2.7, Merck, Darmstadt, Germany) for 10 min at room temperature. The sample was removed and neutralized by adding 1/10 volume of 1M Tris (pH 8.8, Carl Roth, Karlsruhe, Germany).

In the same manner as in the manual procedure, 0.25 µg of antibody targeting CD9, a mix of CD63 or CD81, and an isotype control, respectively, were added per µL of initial Streptavidin MyOne T1 Dynabeads suspension (100 µL) and incubated for 1 h at room temperature under constant agitation for the automated protocol. After four washing steps, 5 mL of healthy cell-free female urine samples (centrifuged at 2000× *g* for 10 min and then at 10,000× *g* for 30 min) with 0.1% (*v*/*v*) Tween 20 were subsequently incubated in 6 mL tubes of the FluidX series (Brooks Life Sciences, Chelmsford, MA, USA) with the antibody-coupled magnetic beads in the IsoMAG-ONE10.0 device for 30 min under constant mixing (at a pipetting speed of 5 mL/min). In the following pipetting steps performed by the device, the beads were washed twice in 4 mL of PBST in two different 6 mL FluidX tubes and once in 1 mL of PBST in a 2 mL FluidX tube, before finally being transferred into 200 µL of PBST in a second 2 mL FluidX tube. The separation and transfer of the beads from one tube to the next were achieved by automated attachment of a magnet to the outside of a pipette tip for 2 min. To compare the automated with the manual isolation procedure of general uEV populations, the procedure by hand was performed using the same antibody-bead preparation, sample volume, and individual. The difference, however, was that the washing time was reduced to 1 min of shaking the tube by hand in 1 mL of PBST due to practical reasons.

### 2.3. IsoMAG-ONE10.0 Device and Protocols

The IsoMAG IMS unit was used as the basis for the development of the IsoMAG-ONE10.0 device [20]. The development was focused on the handling of big- and medium-sized volumes by adding a 10 mL and 1 mL pipet module and the opportunity to introduce different sizes of tubes. The flexibility of the system was enhanced by introducing a rack holder placed on an x-y-movable table and with software that enables the easy setup of new isolation protocols. A general protocol was optimized to specifically isolate EVs from human samples, as shown in Appendix A. To find the optimal parameters for the automated isolation protocol, human cell lines were used to realize a high yield of specific binding and the isolation of relevant biomolecules, as well as to reduce bead loss (see Appendix A).

### 2.4. Nanoparticle Tracking Analysis (NTA)

NTA was performed by the ZetaView TWIN (Particle Metrix, Inning am Ammersee, Germany) using the software ZetaVIEW (Particle Metrix, version 8.05.14 SP7) and a temperature control to maintain constant 25 °C. The flow chamber was flushed with 10 mL PBS between each measurement. The samples were diluted in PBS in an appropriate ratio and measured at 11 positions with the following settings: camera sensitivity of 80 and shutter set to 100. Particle concentrations were calculated by multiplying with the used dilution factor. The samples were measured once on the day that they were collected. They were subsequently frozen, kept at −80 °C for 7 days, thawed, and then measured for a second time.

### 2.5. Western Blotting

For protein separation, hand-casted gels consisting of a separation gel of 12% acrylamide (Carl Roth) and a stacking gel of 5% acrylamide were used. Samples were prepared with 6x Laemmli buffer (0.375 M tris/HCl (Carl Roth); 0.6 M DTT (Carl Roth); 60% (*v*/*v*) glycerol (Merck); 12% (*w*/*v*) SDS (Carl Roth); 0.06% (*w*/*v*) bromophenol blue (Carl Roth)), and heated for 5 min at 90 °C prior to electrophoresis. Protein separation according to their size was performed at 80 V for 20 min, followed by 140 V for 100 min in Towbin buffer (25 mM tris/base (Carl Roth); 192 mM glycine (Carl Roth); pH 8.6) in a Mini-Protean^®^ Tetra Vertical Electrophoresis Cell (BioRad Laboratories, Hercules, CA, USA). After electrophoresis, the gel was washed twice with ddH_2_O and equilibrated in a cathode buffer (48 mM tris/base; 39 mM glycine; 20% (*v*/*v*) methanol (Carl Roth); 0.05% (*w*/*v*) SDS) for membrane transfer. On a Trans-Blot^®^ SD semidry transfer cell (BioRad Laboratories), a sandwich of extra-thick Whatmann paper and a nitrocellulose membrane (0.2 µm, GE, Boston, MA, USA), both soaked in anode buffer (48 mM tris/base; 39 mM glycine; 30% (*v*/*v*) methanol), were placed, followed by the equilibrated gel and another extra-thick Whatmann paper soaked in cathode buffer. Transfer of the proteins from the gel to the membrane was achieved at 25 V for 45 min. The membrane was subsequently blocked in 5% (*w*/*v*) BSA (Carl Roth) in PBST for 1 h at room temperature, and the primary unlabeled antibody (1:1000) was added overnight at 4 °C under constant agitation. After three washing steps of 5 min in PBST at room temperature, incubation with the secondary HRP-conjugated antibody (1:10,000) for 1 h at room temperature followed. Two washing steps of 5 min each with PBST and PBS were conducted before the Clarity Western ECL substrate (Bio-Rad Laboratories) was applied and signals were analyzed in the ChemiDoc MP (Bio-Rad Laboratories). Analyzing band intensities was carried out with Image Lab Software (Bio-Rad Laboratories) using CD9 bands from automated samples as a relative reference.

### 2.6. RNA Isolation and RT-qPCR

After adjusting the volume of urine to creatine and immunomagnetic isolation of uEVs, magnetic beads dissolved in 100 μL PBS were lysed by applying 900 μL TRIzol™ reagent (Thermo Fisher Scientific) for 5 min at room temperature. RNA was isolated by adding 200 µL Chloroform (Carl Roth), followed by 15 s vigorously shaking, 3 min incubation at room temperature, and centrifugation for 15 min at 12,000× *g* at 4 °C. The upper phase was transferred to a new tube and RNA was isolated using the PureLink™ RNA Mini kit (Thermo Fisher Scientific) according to the manufacturer’s instructions. After RNA elution, the volume was adjusted to 175 μL using nuclease-free H_2_O and 18 μL of 3 M (*w*/*v*) NaCl (Carl Roth) was added and mixed by repetitive pipetting. An amount of 600 μL ice-cold 100% ethanol was added and vortexed for 3 sec on maximum speed. The tubes were stored at −80 °C overnight. Precipitated RNA was pelleted using a 30 min centrifugation step at 13,000× *g* and 4 °C. The pellet was washed twice with 900 μL ice-cold 70% (*v*/*v*) ethanol, vortexed for 3 sec at maximum speed, and centrifuged for 5 min at 13,000× *g* and 4 °C. The supernatant was removed, and complete drying of the pellet was allowed for 30 min at 40 °C prior resuspension in nuclease-free H_2_O. RNA analysis was performed by a one-step RT-qPCR reaction using the SCRIPT RT-PCR kit (Jena Bioscience, Jena, Germany). The reaction was prepared on ice in a LightCycler^®^ 480 multi-well plate 96 (Roche) in technical *duplicates* containing 10 μL SCRIPT RT-qPCR Probes-Master, 1 μL 20x EvaGreen (25 μM, Biotium, Fremont, CA, USA), 2 μL primer set (Qiagen, Hilden, Germany, LNA hsa-miR16-5p 5′-UAGCAGCACGUAAAUAUUGGCG-3′), and the purified RNA filled with PCR-grade H_2_O to 20 μL. Then, the following program was conducted in a LightCycler^®^ 480 Instrument II (Roche): 1 cycle of reverse transcription for 10 min at 50 °C, 1 cycle of initial denaturation for 5 min at 95 °C, and 45 cycles of denaturation for 15 sec at 94 °C, and annealing/elongation for 1 min at 56 °C. Melting curves were analyzed for product specificity. Cp values were calculated based on the 2nd derivate calculation for each reaction using the LightCycler^®^ 480 software (Roche). The average of each sample type was calculated in Excel (Microsoft, Redmond, WA, USA, version 1808).

### 2.7. Antibody Microarray

In order to characterize the uEV surface markers, the antibody microarray was designed to accomplish multiplexed analysis in a high-throughput manner. The sciFLEXARRAYER S3 noncontact dispensing system (Scienion, Berlin, Germany) with a PDC70-Type 3 capillary was employed to create reliable and very small, defined spots based on the piezo dispensing capillary technology. On the 3D-Epoxy-Polymer-coated glass slides (PolyAN, Berlin Germany), the direct immobilization of the antibody (see Table 1) was carried out at room temperature and a relative humidity of 75–78% using 0.22 µm filtered dispensing solutions. An antibody concentration of 100 µg/mL in 0.22 µm filtered PBS (Gibco) and the addition of 2.5% glycerol (Carl Roth) were used to create 2.5 nL antibody droplets. After antibody spotting, the slides were kept in the spotting chamber overnight to allow for sufficient immobilization. After at least 16 h, the slides were allowed to dry for 1 h at room temperature prior to the assembly of a slide with a ProPlate^®^ Multi-Well Chamber (Grace Bio-Labs, Bend, OR, USA) to provide cavities to test 16 samples of up to 200 µL sample volume. Next, the slides were incubated with blocking solution (PolyAN) for 1 h at room temperature at 450 RPM before the cell-free urine samples, normalized to 100 mmol/µL creatinine in the final 200 µL PBST, were incubated for 1 h at room temperature under constant agitation. From this step onwards, all incubation and washing steps were performed at room temperature and under constant agitation. After three washing steps of 100 µL PBST for 5 min, the detection antibody mix was added and incubated for 1 h. The mix consisted of APC-conjugated antibodies against human CD9 of 0.03 µg and against human CD63 and CD81 of 1 µg each, in a final volume of 100 µL PBST.

Following three PBST washes of 5 min each and one washing step with MilliQ H_2_O, the slide was dried under a nitrogen gas stream. The slide was then scanned in the GenePix 4200A microarray reader with a gain set to 400 and a power set to 90%. The generated *.GAL file from the spotting process was used for analysis in the GenePix Pro software (Molecular Devices, San José, CA, USA). The mean was calculated in Excel (Microsoft), with blank or isotype control subtracted and log2 calculated. PBST served as control. In addition, antibody immobilization was surveilled by applying a Cy3-labelled anti-mouse antibody (1:5000) on the antibody spots for 1 h at RT.

### 2.8. Transmission Electron Microscopy (TEM)

Visualization of the uEVs in cell-free urine was performed by TEM using an EM 900 (Zeiss Microscopy GmbH, Jena, Germany) at 80 kV. Each sample preparation step on formvar-coated electron microscopy copper grids (200 mesh, Plano, Wetzlar, Germany) was performed for 1 min at RT. First, a 3 μL sample was pipetted on the grid. After three washing steps with H_2_O, a final incubation step with 2% uranyl acetate in H_2_O was performed. Excess liquid was removed, and drying was allowed prior to imaging.

### 2.9. Clinical Samples

A cohort of 26 PCa patients and 16 benign male individuals provided plasma and urine samples, collected at the University Hospital in Mannheim. The participants provided written informed consent and collection of samples according to the ethical standards was approved by the Ethical Committee of the University Hospital in Mannheim (2015-549N-MA). The age of the cohort subjects was between 53 and 78 years. PSA determined in serum ranged from 0.93 to 34.5 ng/mL, and the Gleason score was assigned after a biopsy from the PCa patients was detected between 6 and 9 (see Appendix A). Urine samples from young healthy volunteers were used as controls. Urine was collected from 10 healthy volunteers (5 male, 5 female; age range 22–36 years, see Appendix A) in order to assess size distribution differences in uEVs from cell-free unfiltered urine and filtered urine. Informed consent was obtained from all subjects involved in the study.

The samples were collected and processed to remove cellular components by centrifugation for 10 min at 2000× *g* at RT. The supernatant was transferred into a 50 mL Falcon tube and used fresh or stored at −80 °C. Urine creatinine levels were quantified using the CREJ2 Creatinine Jaffé Gen.2 kit by the automated Roche/Hitachi cobas c311 analyzer (Roche, Basel, Switzerland), as described in the manual. Filtration of cell-free urine was carried out with a cellulose acetate filter membrane with a pore size of 0.2 µm (Roth, Syringe filters ROTILABO^®^ Cellulose acetate) and with a pore size of 0.8 µm (Sartorius, Göttingen, Germany, Minisart^®^ Syringe Filter, Surfactant-free Cellulose Acetate).

### 2.10. Statistics

Data visualization and analysis were carried out with GraphPad Prism 9 (GraphPad Software, San Diego, CA, USA). Diagrams are presented as mean ± SD. Statistical tests were performed using ANOVA with a *p*-value equal to ≤0.05 *, ≤0.01 *, ≤0.001 ***, and ≤0.0001 ****, indicated in the figure legends. Microarray results were prepared as heat maps using the software OriginPro 2020 (OriginLab Corporation, Northampton, MA, USA) after log2 transformation in Excel (Microsoft).

## 3. Results

### 3.1. Size Distribution and Particle Concentration of uEVs from Young Healthy Controls

In order to determine whether the cell-free urine samples would need to be filtered before further uEV processing in a clinical setting, we assessed the size distribution of the uEVs (cell-free unfiltered, filtered at 0.2 µm, and 0.8 µm pore size) from healthy individuals. Regardless of the treatment, a particle size distribution between 20 and 1000 nm was observed in each sample, though particles above 500 nm were rarely detected (Figure 1A). Freezing the samples for one week at −80 °C had no major impact on the particle size and distribution (Figure 1B). The majority of the particles were between 80 and 160 nm in diameter. Interestingly, the filtration step neither resulted in significant changes in the size distribution of the obtained particles nor for the used 0.8 and 0.2 μm pore size filters (Figure 1C). Yet, the particle concentration within the urine samples from the different individuals was highly variable, as expected. Quantities from 9E × 10^9^ o 9E × 10^13^ particles per mL of urine were detected. Normalization to creatinine still resulted in a wide variation between the individuals, but no significant difference was observed between the unfiltered and filtered samples. Similar to their particle size distribution, there was no significant storage effect on the measured particle concentration between the freshly measured and the stored samples (Figure 1D).

Visualization of the uEVs in cell-free urine by TEM showed that the majority of the uEVs was in the size of 100 nm, which is in accordance with the NTA observations. A few large EVs were also detected. Furthermore, as the uEVs were not further isolated, and the cell-free urine was analyzed directly, a high level of background based on urinary proteins and aggregates was observed (Figure 1E,F).

### 3.2. Specific uEV Isolation using Immunomagnetic Beads

Based on the fact that all cells release EVs to mediate intercellular communication, we attempted to isolate a PCa-specific uEV population using the surface protein PSMA. Comparing female and male urine with intermediate PCa risk, Western blot bands for the luminal EV marker TSG101 and the surface EV marker CD9, as well as a faint band for the PSMA were only detected in the PSMA-targeted uEVs from male urine, but not in female urine. CD9-captured uEVs were used as control and showed TSG101 and CD9 signals in similar intensity in both male and female uEVs. Stronger band intensities were observed for CD9-positive uEVs in comparison to PSMA-positive uEVs. The IC control was negative for all tested proteins (Figure 2A). 

Similarly, uEV cargo was analyzed by RT-qPCR for miR-16-5p, because it was reported as a possible reference gene for uEVs, and thus should be present in all uEVs [21]. Using the isotype control, no signal was detected for miR-16-5p, while varying Cp values were observed when capturing CD9-, CD63-, or PSMA-positive uEVs, ranging from Cp values of 37.2 to 39.1 and not detected, respectively. The lowest Cp values, indicating the highest miR-16-5p quantity, were found for the CD63-positive uEVs of patient benign 9 and the PSMA-positive uEVs of patient benign 10, and intermediate PCa risk 11 and 13 (Figure 2B). The PSMA-positive uEVs of benign 9 and intermediate PCa risk showed low to no miR-16-5p expression, which is in accordance with the PSMA expression observed in the antibody microarray (see Section 3.3).

Another aspect of the study was to investigate whether the immunomagnetic isolation of the uEVs could be automated to increase the robustness of the purification technique. Adjustments to the IsoMAG-ONE10.0 device and the protocol optimization for uEVs (see Appendix A) led to a successful uEV isolation. Western blot analysis of CD9, as well as TSG101, of CD9-positive uEVs captured from the same initial urine volume and individual showed higher band intensities for the automated versus the manually performed procedure. Furthermore, the isotype control sample demonstrated reduced amounts of nonspecific binding in the automated sample in comparison to the manually executed sample (Figure 2C).

In general, the total yield of the uEVs isolated with the IsoMAG-ONE10.0 (Figure 3A) was significantly higher than the total yield of the uEVs isolated manually and assessed by Western blot analysis. With the manually isolated uEVs, the yield on average corresponded only to one-third of the uEV yield obtained by the automated procedure (Figure 3D). This could be shown for the CD9-positive populations, as well as the CD63- and CD81-positive uEVs (see Appendix A). The automation of the uEV isolation, furthermore, has proven to hold a significant advantage over the manual isolation process when taking into consideration the treatment of large sample volumes, as is generally the case for urine analysis. With a handling volume of up to 10 mL, the automation provided good mixing qualities for incubation (Figure 3B) and bead washing while retaining magnetic beads using a strong magnet (Figure 3C). Thereby, it reduced not only the hands-on time, but also an undefined amount of magnetic bead loss with the bound uEVs due to imprecision in pipetting or other human error during the manual isolation procedure (Figure 3D).

### 3.3. uEV Surface Protein Analysis of A PCa Cohort

To examine a cohort of 42 patients, Western blot analysis is unsuitable due to its low throughput and time-consuming preparation. Therefore, we used an in-house spotted antibody microarray to study 12 different surface proteins on the uEVs. In line with expectations, the most commonly expressed proteins across the tested PCa cohort were the tetraspanins CD9, CD63, and CD81, generally used as EV markers. CD9 was positive for each individual, whereas CD63 was detected in all samples except for one high-risk PCa patient. On the contrary, CD81 was reduced to only 60%, being positive across the different patients, with the least frequency detected in the benign and high-risk PCa group. Another surface protein expressed frequently on the uEVs, in 75% of all samples, was CD24. Proteins used as cancer markers, such as ROR1, EpCAM, or CA72-4, were observed on the uEVs. The prevalence of ROR1 and EpCAM decreased from benign controls to increasing risk within the PCa patients. High-risk PCa patients were negative for ROR1. Similarly, 38% of the benign group showed signals for EpCAM, decreased with PCa to 14%, while the low-risk PCa group was absent from EpCAM as well as ROR1. CA72-4 was positive for 55% of the subjects without showing a relation to the PCa risk. No difference in the expression intensity was observed in the dependency of the PCa risk groups.

The main interest of the study was to investigate if PSMA is present on the uEVs from PCa patients so to provide a surface target for disease-specific EV isolation from urine. Surprisingly, PSMA expression was observed in less than 17% of the cohort patients, with the most prevalence in the benign and intermediate-risk PCa patients. The absence of PSMA expression, as observed during immunomagnetic isolation from intermediate PCa risk patients 3 and 15 (see Appendix A), was confirmed by the antibody microarray. Furthermore, it has been found that the two different PSMA antibodies, clone LNI-17B (BioLegend) and clone GCP-05 (Thermo Fisher Scientific), showed different frequencies and results within the same sample. PSMA would have been expected to show a higher frequency and expression with more advanced PCa. However, no stratification of PCa patients based on the PSMA expression was observed within the investigated cohort (Figure 4).

## 4. Discussion

Urine might be a powerful liquid biopsy sample for PCa diagnosis based on noninvasively obtained information from uEVs. For the sample processing, we first investigated whether the filtration of cell-free urine influences the particle number and size distribution. Filtration, independent of the pore size of 0.2 or 0.8 µm, had no significant effect on the particle size distribution, indicating that the majority of particles measured correspond to the smaller-sized EVs. Consequently, filtration as a prestep for the purification of the uEVs might be unnecessary and could be eliminated, especially when applying a preferably simple and fast protocol, as favored in clinical settings. As clinical samples are not always used fresh, we analyzed whether the size and number of the uEVs would change between the point when the fresh samples were measured for the first time and when they were measured a second time after storage at −80 °C for one week. When comparing the data obtained from the two measuring time points, no significant changes in the uEV size and number were observed. However, in this study, the storage duration was only one week. Longer time periods at −80 °C might nonetheless have effects on the size and amount of uEVs. Gelibter et al. found a significant increase in the particle size while the particle concentration was reduced when isolated blood plasma EVs were stored at −80 °C for 6 months. On the contrary, when blood plasma was stored under the same conditions and the EVs were isolated after thawing, the changes were less severe, indicating a possible protective effect by the biofluid [22]. In addition, it was shown that the EVs in urine were found to be relatively resistant (up to 18 h at 37 °C) toward endogenous proteolytic activity [23]. 

Although urine has a considerably lower amount of nonvesicular particles, such as lipoproteins and proteins, compared to blood plasma, the isolation of EVs from urine seems to be essential [24]. One study compared the proteomic profile of UC-purified urine with untreated urine by LC-MS and identified a large number of EV proteins that would have likely remained undetected without EV isolation due to the high background of proteins [25]. With an average of 33 µg protein/mL, the majority of urinary proteins are serum albumin, Tamm–Horsfall proteins (uromodulin), aquaporin−1 and −2, uroplakin, and lipoproteins [26], and only ~3% of the total protein accounts to the uEV proteins [2]. This is in accordance with the TEM images obtained from the cell-free urine, showing a high background of nonvesicular structures. It implies the importance of isolation procedures to obtain pure EV samples. Immunomagnetic isolation has therefore been found to be a suitable method because a high purity can be obtained [27]. We were able to show earlier that PSMA-positive cells release PSMA-positive EVs in cell culture, and its feasibility to purify PSMA-positive EVs using immunomagnetic beads [28]. However, bodily fluids are more complex than cell culture supernatants. Based on the fact that all cells release EVs for cell–cell communication, it might be even more important to obtain disease-specific EV subpopulations with a sufficient quantity for analysis, as pathological signatures from PCa-derived uEVs might get lost when analyzing the entire uEV population. Comparing the immunomagnetic isolation from female and male urine, we were able to show that only male urine provided Western blot signals from the PSMA-positive uEVs. While using the same starting volume of urine, differences in CD9, as well as TSG101 band intensities in Western blot, were found between the CD9- and PSMA-positive uEVs captured with immunomagnetic beads from male urine. This indicates a PSMA-positive subgroup of uEVs that is most likely to be present in a lower quantity in urine in comparison to CD9-positive uEVs. In addition, TSG101 signals were more intense in comparison to CD9 signals in the PSMA-positive uEVs when compared to the CD9-positive uEVs, suggesting a possible different cargo sorting. Furthermore, the miRNA analysis of the CD9- and PSMA-positive uEVs by RT-qPCR detected miR-16-5p in some samples, supporting the differences in cargo sorting to function in cellular communication. The absence of miR-16-5p, as well as the analyzed proteins TSG101 and CD9 in the isotype control samples, indicate the high purity of immunomagnetic isolation. The low signals of the protein bands, around 50 kDa, were detected in all uEV samples, indicating an association of the urinary proteins with the uEVs, as shown for the cell-culture-derived EVs that are surrounded by a protein corona after contact with blood plasma proteins [29]. Using the automated procedure in comparison to the manually performed isolation, the process was shown to reduce the hands-on time and to work more efficiently due to the reduced bead loss, resulting in an increased yield, as detected by the protein analysis in the Western blot. Additionally, the preconjugation of the target antibody with the beads could decrease the additional assay time. The device enhances the freedom for downstream processing and analysis. Supposably, a two-step immunomagnetic isolation would be possible in which a specific EV cargo is isolated using magnetic beads after capture and the lysis of the EV population of interest [30]. The independence from the user is a crucial factor, as the automated procedures were able to provide less variation in preparation in comparison to the operator-performed protocols [31]. In general, one drawback of immunomagnetic isolation is the need for a suitable surface target and the availability of a capture molecule. Therefore, tissue- or disease-specific surface molecules are required to be present on the surface of the EVs of interest, and the latter need to be accessible in bodily fluids. For diagnostic applications, the amount of disease-specific EVs might already provide sufficient evidence of the patient status. Nevertheless, it seems to be more important to obtain additional information based on the analysis of EV cargo, which is influenced by the cell type and (patho-)physiological stimuli [3,5]. Thus, the stratification of patients might be possible to distinguish the indolent from aggressive PCas, or to assess the heterogeneity and dynamics of the tumor. As conventional tissue biopsy reflects only a portion of the tumor at a single time point, while being irreproducible in most cases, liquid biopsy using EVs could solve this issue. To our knowledge, a combination of disease-specific enrichment of the EVs and cargo analysis is still under investigation in the research and needs further development before being implemented in clinics.

To identify suitable disease-related targets, microarrays have shown an enormous diagnostic potential due to their high throughput and the ability to detect nucleic acids, proteins, or lectins [32,33]. In this investigation, the antibody microarray was a beneficial technique to identify the uEV surface proteins from the cell-free urine normalized to creatinine. The use of normalization is of utmost importance based on the fact that urine is a sample material highly variable in comparison to, e.g., blood plasma. Depending on the patient’s individual intake of fluids, urine ranges from a very diluted to a very concentrated bodily fluid. Creatinine is a suitable normalizer due to its relatively constant secretion within and across individuals. Nevertheless, it has to be considered that, in some cases, creatinine is not applicable, e.g., in kidney diseases [34]. Ubiquitous EV surface markers, CD9, CD63, and CD81, were detected more often than specific markers. Although CD63 and CD9 were found only in approximately two-thirds and CD81 was present in all analyzed 60 different cell lines analyzed by LC-MS/MS [35], the microarray showed results vice versa. CD9 was the most frequently detected marker, followed by CD63. CD81 was less often present, within only 57% of the samples. One explanation might be the more complex sample, as the urine contains EVs from a multitude of different cells rather than one type of cell grown under laboratory conditions. Another reason could be the effect of the disease changing the expression pattern, and consequently the cargo of the released EVs. A relatively high abundance of CD24 in 75% of samples was found. CD24 is expressed by tubule cells and podocytes in the kidney, and is therefore released frequently into the urine, and has subsequently been suggested as a potential uEV marker [36]. However, the requirements for a suitable EV marker necessitate the presence on all EVs, which was not the case in this study. Similarly, cancer-related markers, such as ROR1 or EpCAM, which are overexpressed in PCa, but also in other carcinomas such as colon cancer, were not able to distinguish between the benign and malignant changes in the prostate [37]. In regard to PCa diagnosis, the microarray results could not identify a vesicular PSMA presence associated with PCa. Certainly, the expression is not restricted to one type of tissue. PSMA is found in secretory glands, ovary, breast, liver, small intestine, and kidney [38,39]. However, the current use of PSMA as a target for the radioisotope-labeled imaging of PCa cells is particularly advantageous due to the high expression in all PCa stages and the overexpression in androgen-independent or metastatic cells [40]. Furthermore, PSMA antibody-drug conjugates for the PCa treatment are currently being investigated in clinical trials [41]. This cellular PSMA presence does not match with the findings for the uEVs, where only less than 17% of the total analyzed cohort was PSMA-positive. Moreover, PSMA could not be identified as a suitable EV biomarker due to its varying profile in benign- and PCa-derived uEVs with no distinct profile in dependency of the stage of the disease. One reason might be that the used antibodies were unable to detect the different isoforms of PSMA or glycosylation patterns on the uEVs.

## 5. Conclusions

In recent years, EVs have attracted interest due to mediating intercellular communication, and thus providing diagnostic information by disease-specific signatures from liquid biopsy samples. The size distribution and particle concentration of cell-free urine allow for the conclusion that the samples collected from healthy individuals do not necessitate special filtration treatment prior to the isolation of the uEVs. However, it needs to be taken into consideration that this might not be the case for patients with a clinical record. In addition, urine is a highly variable biofluid and necessitates normalization strategies applicable for the uEVs.

Although PSMA was not identified as a perfect EV surface protein for PCa, we were able to show that specific uEV immunomagnetic isolation directly from cell-free urine was possible and more effective when using automated procedures. Despite the ability of EVs to mirror the cell of origin, a discrepancy between the cells and EVs seems to be present in some cases. This reinforces the imperative to screen for disease-specific signatures on and in the EVs, as they potentially differ in comparison to their cellular counterpart. Simple, fast, and high-throughput methods are therefore inevitable, which we were able to show with the antibody microarray to analyze up to 33 EV surface proteins per sample *in triplicate*, devoid of uEV isolation. In the future, it might pinpoint potential surface targets for the immunomagnetic isolation of EVs and provide a tool for early cancer diagnosis from liquid biopsy samples.

## Figures and Tables

**Figure 1 cancers-14-02987-f001:**
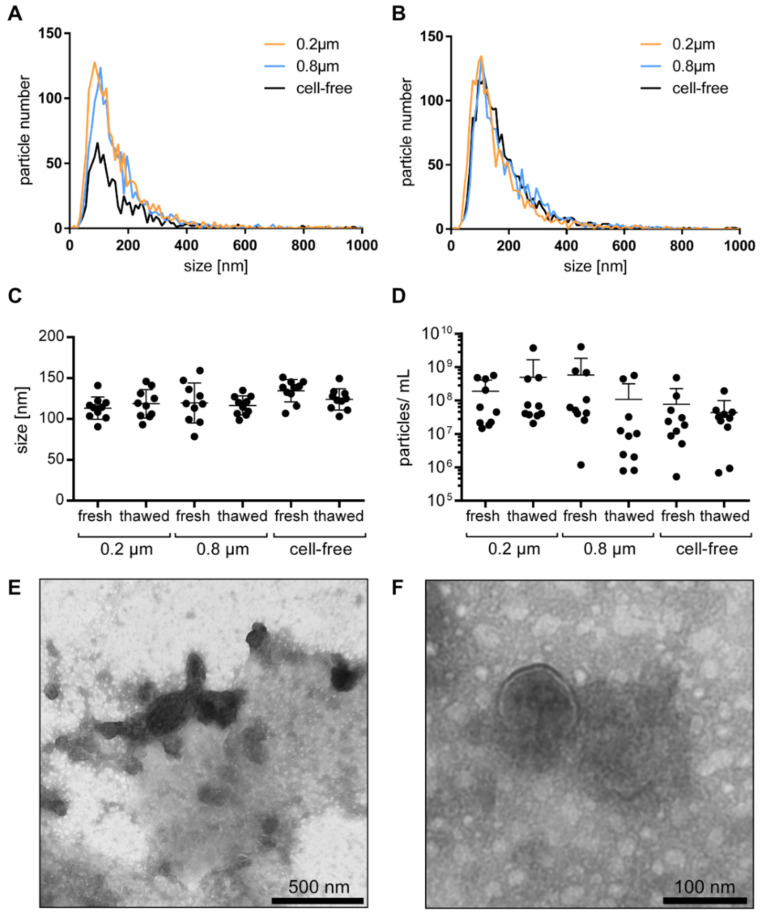
The uEV size, size distribution, and particle concentration: uEV size distribution of nonfiltered, 0.2 µm, and 0.8 µm filtered cell-free urine from healthy individuals analyzed by NTA fresh (**A**) and after one week of storage (**B**). Particle size (**C**) and particle number per mL of urine normalized to creatinine (**D**) from fresh and thawed nonfiltered, 0.2 µm, and 0.8 µm filtered cell-free urine by NTA, shown as mean ± SD. Visualization of uEVs in cell-free urine by TEM in wide-field with scale bar representing 500 nm (**E**) and in near-field with scale bar representing 100 nm (**F**).

**Figure 2 cancers-14-02987-f002:**
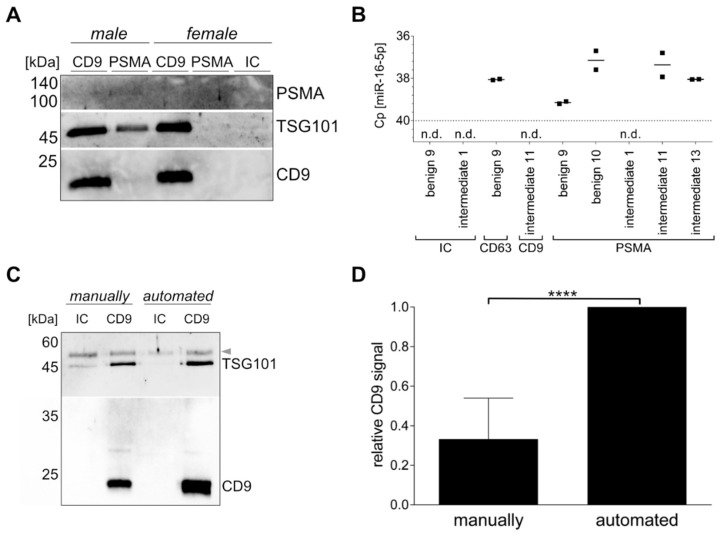
Specific uEV isolation using immunomagnetic beads. Western blot analysis of CD9- or PSMA-positive uEVs and the isotype control (IC) from female and male intermediate PCa risk urine for PSMA, TSG101, and CD9 (**A**). RT-qPCR analysis of miR-16-5p of CD9-, CD63-, or PSMA-positive uEVs and the isotype control (IC). n.d. = not detected. (**B**). Comparison of manual and automated procedure analyzed in Western blot for TSG101 and CD9 of CD9-positive uEVs and the isotype control (IC) from healthy female urine. Arrow indicates urinary proteins (**C**). Ubiquitous captured uEVs based on CD9 or a mix of CD61 and CD81 analyzed by Western blot by CD9 summarized for n = 5 ± SD, **** *p* ≤ 0.0001 (**D**).

**Figure 3 cancers-14-02987-f003:**
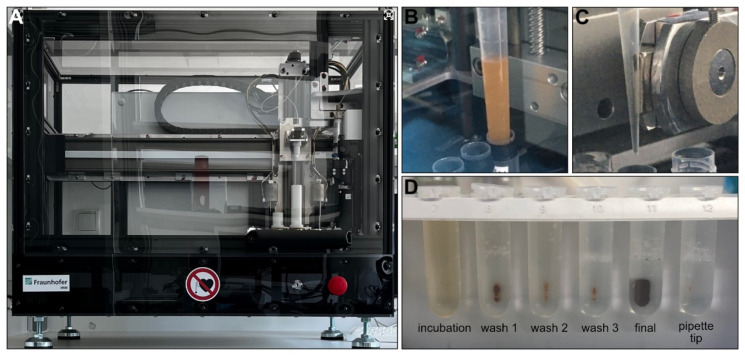
IsoMAG-ONE10.0 device used for immunomagnetic isolation of tetraspanin-positive EV populations from urine (**A**). Sample incubation with antibody-coated beads under constant mixing (**B**). Retaining antibody-coated beads with captured EVs from the sample using a strong magnet automatically moved directly to the outside of the tip (**C**). FluidX tube contents were assessed for bead loss in the magnetic rack after assay completion (**D**).

**Figure 4 cancers-14-02987-f004:**
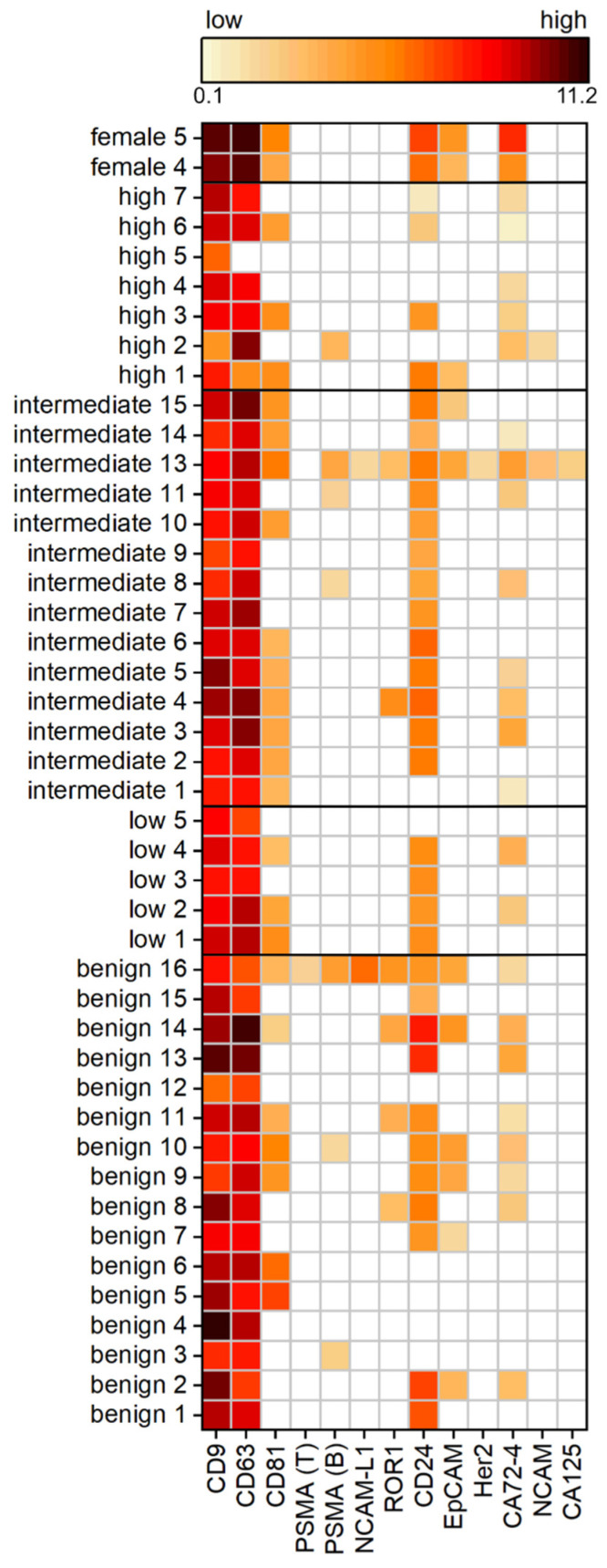
Expression analysis of uEV surface proteins of a PCa cohort with benign patients and patients with low, intermediate, and high risk according to the Gleason score, as well as healthy female controls by antibody microarray normalized to creatinine. The heat map displays log2 calculated background-subtracted values with no (white) and low (light orange) to high expression (dark red), n = 3.

## Data Availability

Datasets used and/or analyzed during the current study are available from the corresponding author.

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
