# Peer review of "Prostate-Specific Membrane Antigen (PSMA)-Positive Extracellular Vesicles in Urine—A Potential Liquid Biopsy Strategy for Prostate Cancer Diagnosis?"

_cancers, 2022, doi:10.3390/cancers14122987_

Round 1

Reviewer 1 Report

As the cargo of biological molecules reflects the patients’ situation, exosomes can be used as new diagnostic tools to serve as a non-invasive liquid biopsy. Supplementary research and development are mandatory for clinical applications, but the clinical utility of exosomes is promising. 

Thus, the study addresses a very timely and important topic in prostate cancer research.

Some changes are suggested:

-       The medical treatment scenario for prostate cancer should be further discussed, and some recent paper added, only for a matter of consistency (PMID: 33557050 ; PMID: 32911806)

-       A linguistic revision is necessary, since there are some little grammar mistakes

-       The limitations of the current study, and especially the possible impact on everyday clinical practice, should be further discussed.

We suggest major changes.

Author Response

Dear Reviewer,

We thank you for your comments and suggestions about our manuscript „Prostate-specific membrane antigen (PSMA) -positive extracellular vesicles in urine - a potential liquid biopsy strategy for prostate cancer diagnosis?“ (Submission ID cancers-1774050) authored by Susann Allelein, Keshia Aerchlimann, Gundula Rösch, Roxana Khajehamiri, Andreas Kölsch, Christian Freese, and Dirk Kuhlmeier.

We addressed all your comments and submitted a revised version of our paper. Here is a detailed response to your comments:

R1-1: The medical treatment scenario for prostate cancer should be further discussed, and some recent papers added, only for a matter of consistency (PMID: 33557050; PMID: 32911806)

We thank you for this suggestion. In the discussion section, we added the fact that PSMA is investigated as a target for treatment in clinical trials. Based on our focus on early diagnosis we would not like to go much further into detail regarding prostate cancer therapy.

Changes to the manuscript:

…However, the current use of PSMA as a target for radioisotope labeled imaging of PCa cells is in particular advantageous due to the high expression in all PCa stages and the over-expression in androgen-independent or metastatic cells [40]. Furthermore, PSMA antibody-drug conjugates for PCa treatment are currently investigated in clinical trials [41]. This cellular PSMA presence does not …

R1-2: A linguistic revision is necessary, since there are some little grammar mistakes

Was performed.

R1-3: The limitations of the current study, and especially the possible impact on everyday clinical practice, should be further discussed.

We included a section in the discussion part.

Changes to the manuscript:

…availability of a capture molecule. Therefore, tissue- or disease-specific surface molecules are required to be present on the surface of the EVs of interest, and the latter need to be accessible in bodily fluids. For diagnostic applications, the amount of disease-specific EVs might already provide sufficient evidence of the patient status. Nevertheless, it seems to be more important to obtain additional information based on the analysis of EV cargo which is influenced by the cell type and (patho-)physiological stimuli [3, 5]. Thus, stratification of patients might be possible to distinguish indolent from aggressive PCa or to assess the heterogeneity and dynamics of the tumor. As conventional tissue biopsy reflects only a portion of the tumor at a single time point, while being irreproducible in most cases, liquid biopsy using EVs, could solve this issue. To our knowledge, a combination of disease-specific enrichment of EVs and cargo analysis is still under investigation in research and needs further development before implementing in clinics.

Reviewer 2 Report

The manuscript by Allelein et.al. explored the possibility to detect PSMA EV from urine samples using a custom protein array as a means for early detection of prostate cancer. They showed isolation of EV from urine using antibody conjugated mag-beads using an automated device and characterized the cargo-load of purified EVs. Although unsuccessful in the original goal, this study could represent an important early step in potentially utilizing EVs as a biomarker. 

Major Comment

1. Figure 2A. Western blot with PSMA antibody does not seems to detect presence of PMSA in EVs purified with PSMA antibody (i.e. no clear bands on the blot). Could the authors comment on this finding? How do the authors know the EVs, if any, truly contain PSMA?

Minor comment

1. Line 320, "unspecific" should be "non-specific"

2. Table S1, the last "benign" sample has GS of 6. Please clarify. Also there is no "7a" in Gleason score.

Author Response

Dear Reviewer,

We thank you for your comments and suggestions about our manuscript „Prostate-specific membrane antigen (PSMA) -positive extracellular vesicles in urine - a potential liquid biopsy strategy for prostate cancer diagnosis?“ (Submission ID cancers-1774050) authored by Susann Allelein, Keshia Aerchlimann, Gundula Rösch, Roxana Khajehamiri, Andreas Kölsch, Christian Freese, and Dirk Kuhlmeier.

We addressed all your comments and submitted a revised version of our paper. Here is a detailed response to your comments:

R2-1: Figure 2A. Western blot with PSMA antibody does not seems to detect presence of PMSA in EVs purified with PSMA antibody (i.e. no clear bands on the blot). Could the authors comment on this finding? How do the authors know the EVs, if any, truly contain PSMA?

The presence of EVs is detected by a combination of a luminal EV protein, TSG101, and a surface protein on EVs, CD9. According to the MISEV guidelines both proteins can be used as a general EV marker. Using immunomagnetic beads for EV isolation, both proteins should be present to show the successful isolation. If only one protein would be detectable it would not be clear whether or not EVs, free proteins, or membrane fragments would have been purified. The PSMA band on the Western Blot is indeed hard to see. We agree that the signal intensity is very low and adjust the contrast of the image. The presence of PSMA in this particular sample was confirmed with the antibody microarray. The low signal intensity could indicate that only a few PSMA molecules are found on PSMA+EVs or that there are only a few PSMA+ EVs in comparison to CD9+ EVs.

R2-2: Line 320, "unspecific" should be non-specific

Thank you. We have changed it in the manuscript.

R2-3: Table S1, the last "benign" sample has GS of 6. Please clarify. Also there is no "7a" in Gleason score.

Patient benign 16 had a high PSA value and underwent needle biopsy which showed a histopathology corresponding to a Gleason Score of 6. However, after the removal of the prostate, no primary tumor was detected.

Changes to the manuscript, supplementary part:

…Gleason Score (GS) determined after prostate tissue needle biopsy.

The Gleason score is determined by histology of the prostate tissue from biopsy and can help to predict the prognosis of the patient. It was observed that it is important to distinguish a Gleason score of 7 (doi: 10.1097/PAS.0000000000000530.). 7a (3+4) has a low aggressive behavior whereas 7b (4+3) has a moderate aggressiveness resulting in a worse prognosis compared to 7a.

Round 2

Reviewer 1 Report

acceptance